# Consistent Nonparametric Density Estimation with Neural Networks: A Unary Classification Approach

**Andrey I. Perminov, Andrey P. Kovalenko, Denis Y. Turdakov**
Research Center for TAI
Ivannikov Institute for System Programming of the Russian Academy of Sciences (ISP RAS)
Moscow, Russia
`{perminov,a.p.kovalenko,turdakov}@ispras.ru`

## Abstract

A consistent nonparametric method for probability density estimation using neural networks is presented. The core idea is to train a multilayer perceptron (MLP) with piecewise linear hidden activations as a unary classifier, distinguishing true data from uniform background noise. The learned classifier's output probability is analytically transformed into a density estimate, guaranteeing statistical reliability based on the theory of unary classification. This framework provides a flexible, neural network-based alternative to classical nonparametric estimators, such as histograms, kernel density methods, and k-nearest neighbors (k-NN). Once trained, the estimator has constant time complexity $O(1)$ per evaluation, independent of the training set size. The proposed estimator is quantitatively evaluated on synthetic distributions with known analytical forms, using Kullback-Leibler (KL) divergence as the principal performance metric. Experimental results demonstrate that the neural density estimator not only enjoys a solid theoretical foundation but also achieves lower KL divergence compared to standard baseline methods, establishing it as a powerful and trustworthy tool for nonparametric density estimation.

## 1 Introduction

Density estimation is a fundamental problem in mathematical statistics, with applications ranging from anomaly detection to scientific data analysis. Classical nonparametric methods – histograms, kernel density estimation (KDE), and k-nearest neighbors (k-NN) – are well-studied but have known limitations: histograms are piecewise constant and bin-sensitive, KDE requires careful bandwidth tuning, and k-NN incurs high computational costs at inference time.

Recent neural approaches offer flexibility but often lack the theoretical guarantees essential for trustworthy AI and safety-critical applications. Many either impose parametric assumptions or provide no consistency guarantees.

This paper introduces a nonparametric neural density estimator that is both statistically consistent and computationally efficient. The key idea is to reformulate density estimation as unary classification: a multilayer perceptron with piecewise linear activations is trained to distinguish data from uniform background noise, and its output is transformed into a density estimate via a simple formula. Consistency follows from recent results connecting piecewise linear MLPs to histogram-based regression (Eliseev et al., 2025) and the theoretical foundations of unary classification (Lukianov et al., 2024), positioning the method as a reliable tool for trustworthy AI.

The main contributions of this work are as follows:

- A consistent nonparametric density estimator via unary classification with MLPs.
- Theoretical consistency established through connection to histogram methods.
- Extensive evaluation on synthetic distributions with known densities.
- Systematic comparison with classical estimators showing superior KL divergence.

- Analysis of computational advantages, including constant-time inference.

The paper is organized as follows. Section 2 reviews related work. Section 3 describes the proposed method and its theoretical foundations. Section 4 details the experimental setup. Section 5 presents the results. Section 6 discusses implications and limitations. Section 7 concludes the paper.

## 2 RELATED WORK

This section reviews existing approaches to density estimation, focusing on classical nonparametric methods and modern techniques directly relevant to the problem.

### 2.1 CLASSICAL NONPARAMETRIC DENSITY ESTIMATION

Histograms represent the most elementary approach to density estimation. The input space is partitioned into bins, and the density within each bin is estimated by the proportion of samples falling into that bin divided by the bin volume. Despite its simplicity, the histogram estimator is consistent under appropriate conditions on bin width (Scott, 2011; Freedman & Diaconis, 1981). However, the resulting estimate is piecewise constant and highly sensitive to the choice of bin boundaries and width.

Kernel density estimation (KDE) addresses the smoothness limitation of histograms by placing a smooth kernel function at each data point and summing their contributions. The KDE estimator takes the form $\hat{f}(\mathbf{x}) = \frac{1}{n} \sum_{i=1}^{n} K_{\mathbf{H}}(\mathbf{x} - \mathbf{x}_i)$, where $K_{\mathbf{H}}$ is a kernel function with bandwidth matrix $\mathbf{H}$. Under mild conditions, KDE is consistent and asymptotically normal (Wand & Jones, 1994; Silverman, 1986). Practical performance, however, depends critically on bandwidth selection. Additionally, KDE requires a meaningful metric in the input space and careful normalization when variables have different scales — a common issue in multivariate settings. The method also becomes inefficient in high dimensions due to the sparsity of data.

The k-nearest neighbors (k-NN) approach offers an alternative that adapts locally to data density. For a given point $\mathbf{x}$, the distance $R_k(\mathbf{x})$ to the $k$-th nearest neighbor is found, and the density is estimated as $\hat{f}(\mathbf{x}) = k/(nV_k(\mathbf{x}))$, where $V_k(\mathbf{x})$ is the volume of the sphere of radius $R_k(\mathbf{x})$. The k-NN estimator is consistent if $k \to \infty$ and $k/n \to 0$ as $n \to \infty$ (Loftsgaarden & Quesenberry, 1965; Bhattacharya & Mack, 1990). Like KDE, k-NN depends on the choice of metric and is sensitive to scaling of individual dimensions. A significant practical drawback is that inference requires storing the entire training set and computing distances to all points, leading to $O(n)$ time per query.

### 2.2 NOISE CONTRASTIVE ESTIMATION

A particularly relevant line of research is Noise Contrastive Estimation (NCE) (Gutmann & Hyvärinen, 2010). NCE trains a binary classifier to distinguish data samples from artificially generated noise samples. The key insight is that the optimal classifier yields an estimate of the data density up to a normalizing constant. NCE has been successfully applied to various problems, including natural language processing (Mikolov et al., 2013) and unsupervised learning (Mnih & Teh, 2012).

The proposed method shares the high-level intuition of NCE but differs in several important aspects. First, NCE assumes a parametric form for the density, while the present approach is fully nonparametric. Second, NCE requires careful selection of the noise distribution and may need multiple noise samples per data point for stability. Third, the theoretical justification provided by (Lukianov et al., 2024; Eliseev et al., 2025) offers a direct connection to classical histogram methods, ensuring consistency without parametric assumptions.

### 2.3 NEURAL NETWORKS FOR DENSITY ESTIMATION

While neural networks have been extensively used for high-dimensional generative modeling, their application to classical density estimation has received comparatively less attention. Some works have explored using neural networks to approximate density functions directly by minimizing a divergence measure between the empirical distribution and the model (Magdon-Ismail & Atiya, 1998;

Bengio & Bengio, 1999; Rothfuss et al., 2019). These approaches often require careful regularization to prevent overfitting and lack theoretical consistency guarantees.

Mixture density networks (Bishop, 1994) model conditional densities but assume a parametric form (Gaussian mixtures) for the output distribution. The method presented in this paper makes no parametric assumptions, leveraging the universal approximation capacity of MLPs to learn arbitrary distributions directly from data.

## 3 PROPOSED METHOD

This section describes the proposed approach to density estimation. First, the problem is formally stated. Then, the unary classification framework is introduced, including the neural network architecture and its theoretical foundation. Finally, the transformation from classifier output to density estimate is derived.

### 3.1 PROBLEM STATEMENT

Let $\{\mathbf{X}_1, \ldots, \mathbf{X}_n \in \mathbb{R}^d\}$ be independent and identically distributed random vectors drawn from an unknown probability distribution with continuous probability density function $f(\mathbf{x})$. The goal is to construct an estimate $\hat{f}(\mathbf{x})$ of the true density based on the observed sample.

It is assumed that all data points lie within a compact set $K = [0, 1]^d$. This assumption is not restrictive, as any bounded dataset can be scaled and translated to fit within the unit hypercube. Moreover, it is required for the theoretical results concerning the approximation capabilities of multilayer perceptrons that underlie the consistency of the proposed method.

### 3.2 UNARY CLASSIFICATION

The proposed method reframes density estimation as a unary classification problem. Consider an augmented dataset consisting of the original data points $\mathbf{X}_i$ and artificially generated noise points $\mathbf{Z}_j$ drawn from the uniform distribution on $K$. Each data point is assigned a label $Y = 1$, and each noise point is assigned a label $Y = 0$. A classifier trained on this augmented dataset learns to distinguish between regions where data are present and regions where only noise exists.

The theoretical foundation for this approach rests on two recent results. In (Lukianov et al., 2024), it is shown that adding background noise to a binary classification problem preserves the behavior of the Bayesian classifier on the support of the data distribution, while forcing the classifier to output the neutral label 0 in regions where no data are present. Unary classification adapts this idea by discarding the negative class entirely, retaining only the data class (label 1) and the noise class (label 0). The classifier then effectively learns to identify the support of the data distribution.

The classifier is implemented as a multilayer perceptron (MLP) with piecewise linear activation functions in the hidden layers. The choice of piecewise linear activations is crucial for theoretical reasons. In (Eliseev et al., 2025), it is demonstrated that such a network converges to a histogram-based Bayesian classifier under appropriate conditions. The argument proceeds by showing that a piecewise linear MLP induces a hierarchical partitioning of the input space. Within each cell of this partition, the network output approximates the empirical class frequency $n_1/(n_0 + n_1)$, where $n_1$ and $n_0$ are the numbers of data and noise points falling into that cell. This asymptotic equivalence to a histogram estimator guarantees the consistency of the MLP-based unary classifier. Unlike the piecewise constant histogram, however, the MLP produces a piecewise linear approximation, enabling more accurate estimation while retaining the consistency property.

During training, the network is presented with equal numbers of data points and uniformly generated noise points. The objective is to minimize the discrepancy between the network output and the target labels. The training process generates fresh noise samples at each optimization step, ensuring that the network learns to discriminate against the uniform distribution rather than memorizing a fixed set of noise points.

### 3.3 Density Estimation via Unary Classification

The connection between the classifier output and the target density follows from a simple application of Bayes' theorem. Let $p_{\text{data}}(\mathbf{x}) = f(\mathbf{x})$ be the data density and $p_{\text{noise}}(\mathbf{x}) = 1$ be the uniform noise density on $K$ (since the volume of $K$ is 1). If the prior probabilities of data and noise are equal, then:

$$P(Y = 1 \mid \mathbf{x}) = \frac{f(\mathbf{x})}{f(\mathbf{x}) + 1}. \tag{1}$$

Solving for $f(\mathbf{x})$ yields:

$$f(\mathbf{x}) = \frac{P(Y = 1 \mid \mathbf{x})}{1 - P(Y = 1 \mid \mathbf{x})}. \tag{2}$$

Thus, given an estimate $c_n(\mathbf{x})$ of the posterior probability produced by the trained MLP, the density estimate is obtained as:

$$\hat{f}(\mathbf{x}) = \frac{c_n(\mathbf{x})}{1 - c_n(\mathbf{x})}. \tag{3}$$

In practice, the network output is clipped to the interval $[\varepsilon, 1 - \varepsilon]$ with a small $\varepsilon$ to avoid numerical issues when $c_n(\mathbf{x})$ approaches 0 or 1. The value $\varepsilon = 10^{-8}$ is used in the experiments.

The consistency of this density estimator follows directly from the consistency of the underlying classifier. Since $c_n(\mathbf{x})$ converges to the true posterior probability $P(Y = 1 \mid \mathbf{x})$ as the sample size increases, the transformation equation 3 converges to the true density $f(\mathbf{x})$. The uniform noise density is exactly 1 on $K$, so no additional normalization constant is required.

## 4 Experimental Setup

This section describes the experimental design used to evaluate the proposed density estimation method. First, the synthetic distributions employed for evaluation are presented. Then, the neural network architectures and training details are specified. Finally, the baseline methods and evaluation metrics are described.

### 4.1 Synthetic Distributions

The proposed method is evaluated on five two-dimensional distributions with known analytical density functions. For each distribution, 2000 samples are generated for training. All distributions are scaled to lie within the unit square $[0, 1]^2$ to satisfy the compactness assumption required by the theoretical results.

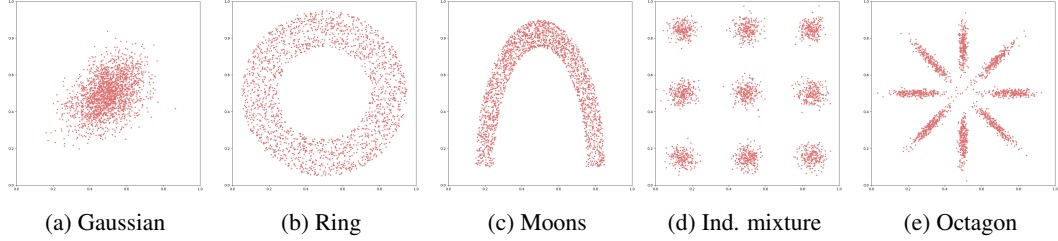

| (a) Gaussian | (b) Ring | (c) Moons | (d) Ind. mixture | (e) Octagon |

Figure 1: Random samples (2000 points) from each synthetic distribution used in the experiments.

- **Gaussian** (Figure 1a): a bivariate normal distribution with mean $\boldsymbol{\mu} = [0.5, 0.5]$ and covariance matrix $\boldsymbol{\Sigma} = \begin{bmatrix} 0.01 & 0.004 \\ 0.004 & 0.01 \end{bmatrix}$. This represents a simple unimodal distribution concentrated near the center of the unit square.

- **Ring** (Figure 1b): points uniformly distributed within a circular ring centered at $(0.5, 0.5)$ with inner radius $0.25$ and outer radius $0.45$. This distribution has a characteristic hole in the middle, presenting a challenge for methods that assume unimodality.

- **Moons** (Figure 1c): a moon shaped uniform distribution (half-circle pointing downward). This distribution exhibits strong nonlinearity and multimodality, making it a common benchmark in machine learning.

- **Independent Gaussian mixture** (Figure 1d): a product of two independent one-dimensional Gaussian mixtures. Each coordinate follows a mixture of three Gaussians with means at $0.15$, $0.5$, and $0.85$, each with variance $0.19^2$ and equal weights. The resulting two-dimensional distribution has nine modes arranged in a grid.

- **Octagon Gaussian mixture** (Figure 1e): a mixture of eight Gaussian distributions arranged in a circular pattern. The mean of the $i$-th component is $(0.5 + 0.25 \cos(\pi i/4), 0.5 + 0.25 \sin(\pi i/4))$. The covariance matrices are rotated to align with the radial direction, with variance $0.16^2$ in the tangential direction and larger variance in the radial direction. This creates a flower-like pattern with eight petals.

In addition to these two-dimensional distributions, experiments are also conducted on three-dimensional and five-dimensional Gaussian distributions to assess the behavior of the method in higher dimensions. For these cases, the covariance matrices are set to $0.01 I_d$, and the means are at $0.5 \cdot \mathbf{1}_d$.

## 4.2 Neural Network Architectures and Training

Five multilayer perceptron architectures with different depths and widths are considered:

- $d$-40-1 (one hidden layer with 40 neurons);
- $d$-20-20-1 (two hidden layers with 20 neurons each);
- $d$-40-40-1 (two hidden layers with 40 neurons each);
- $d$-80-80-1 (two hidden layers with 80 neurons each);
- $d$-20-20-20-1 (three hidden layers with 20 neurons each).

Here, $d$ denotes the input dimension, and the numbers indicate the number of neurons in each hidden layer. All hidden layers use the absolute value function $\text{abs}(x)$ as the activation function. The output layer has a single neuron with no activation function.

During training, each batch consists of 64 data points and 64 freshly generated uniform noise points. Noise points are sampled independently at each optimization step from the uniform distribution on $[0, 1]^d$. The network is trained to minimize the mean squared error between its output and the target labels (1 for data, 0 for noise).

Optimization is performed using the Adam optimizer with a learning rate of $0.004$. $L_2$ regularization with coefficient $0.001$ is applied to all weights. Training proceeds for 100 epochs, with each epoch consisting of iterations over all data points paired with freshly generated noise.

After training, the network output $c_n(\mathbf{x})$ is clipped to the interval $[0, 1 - \varepsilon]$ with $\varepsilon = 10^{-8}$ before applying the transformation equation 3.

## 4.3 Baseline Methods

The proposed method is compared against three classical nonparametric density estimators:

- **Histograms**: the unit square is partitioned into a regular grid of bins. Different numbers of bins per dimension are considered: 5, 10, 20, 30, and 40. The density estimate is constant within each bin and equal to the proportion of data points falling into that bin divided by the bin area.

- **Kernel density estimation (KDE)**: five kernel functions are evaluated: Gaussian, tophat, Epanechnikov, exponential, and cosine. A fixed bandwidth of $0.05$ is used for all kernels.

The density estimate is computed as $\hat{f}(\mathbf{x}) = \frac{1}{n}\sum_{i=1}^{n} K_{\mathbf{H}}(\mathbf{x} - \mathbf{x}_i)$, where $K_{\mathbf{H}}$ is the kernel with bandwidth matrix $\mathbf{H} = h^2 I$ and $h = 0.05$.

- **k-nearest neighbors (k-NN)**: five values of $k$ are evaluated: 5, 10, 20, 50, and 100. For each point $\mathbf{x}$, the distance $R_k(\mathbf{x})$ to the $k$-th nearest neighbor among the training data is found, and the density is estimated as $\hat{f}(\mathbf{x}) = k/(nV_k(\mathbf{x}))$, where $V_k(\mathbf{x})$ is the volume of the sphere of radius $R_k(\mathbf{x})$ in $\mathbb{R}^d$.

For all baseline methods, multiple hyperparameter configurations are evaluated, and all results are reported without preselection of the "best" configuration. This provides a comprehensive view of each method's performance across its hyperparameter space.

## 4.4 EVALUATION METRICS

The quality of the density estimates is assessed by comparing them to the true analytical density functions. Two metrics are used: Mean Squared Error (MSE) and Kullback-Leibler (KL) divergence.

A crucial technical note concerns the application of KL divergence to density estimates. KL divergence is defined for probability distributions, requiring that the estimated function integrate to 1. However, density functions are not confined to the range $[0, 1]$ and their values can exceed 1 in regions of high concentration. Simply normalizing by the sum of estimated densities over a grid is incorrect, as this treats the density as a probability mass function. In this work, KL divergence is computed by numerical integration over the grid, ensuring that both $\hat{f}$ and $f$ are treated as true densities. The issue of improper KL divergence application in some existing works is discussed further in Section 6.

All metrics are evaluated on a regular grid covering $[0, 1]^d$. For two-dimensional distributions, the grid resolution is $500 \times 500$ points (250000 evaluation points). For three-dimensional distributions, a $100 \times 100 \times 100$ grid is used (1000000 points). For five-dimensional distributions, a $50^5$ grid is used (312500000 points), which is the maximum feasible given computational constraints.

## 4.5 COMPUTATIONAL EFFICIENCY

In addition to estimation accuracy, computational efficiency at inference time is evaluated. For each method, the total time required to compute density estimates on the full evaluation grid is recorded. This is particularly relevant for k-NN and KDE methods. The k-NN estimator can be accelerated using tree-based data structures such as KD-trees or ball trees, which in practice often achieve sublinear query times. However, these structures still incur overhead and may degrade to linear complexity in high dimensions or with unfavorable data distributions. KDE with non-compact kernels (such as Gaussian) typically requires summing contributions from all training points for each evaluation point, resulting in $O(n)$ complexity per prediction unless approximations are used. For large evaluation grids, this computational burden becomes substantial. In contrast, the proposed MLP method performs a fixed number of operations per prediction ($O(1)$) regardless of the training set size, enabling efficient inference even on dense grids. Once trained, the network can evaluate new points in constant time, requiring only a forward pass through a fixed number of layers.

## 5 RESULTS

This section presents the experimental results of the proposed density estimation method. First, the performance of different MLP architectures is compared across the synthetic distributions. Then, the proposed method is compared against classical baseline estimators. Finally, visualizations of the estimated densities are provided, and computational efficiency is analyzed.

## 5.1 COMPARISON OF MLP ARCHITECTURES

Six MLP architectures were evaluated on each of the five two-dimensional distributions. Tables 1 and 2 show the KL divergence and mean squared error, multiplied by $10^3$) achieved by each architecture on each distribution. Lower values indicate better agreement with the true density.

Table 1: KL divergence (multiplied by $10^3$) for different MLP architectures across synthetic distributions. Lower is better.

| Architecture | Gaussian | | | Ring | Moons | Ind. mixture | Octagon |
|---|---|---|---|---|---|---|---|
| | 2d | 3d | 5d | | | | |
| d-40-1 | 12.13 | 68.17 | 520 | 297 | 379 | 701 | 913 |
| d-20-20-1 | **8.34** | 34.72 | 318 | 172 | 237 | 366 | 782 |
| d-40-40-1 | 11.99 | 86.12 | 152 | 83.46 | 156 | 153 | 271 |
| d-80-80-1 | 10.42 | 42.45 | **88.54** | **76.07** | 160 | 285 | 445 |
| d-20-20-20-1 | 11.95 | **34.63** | 203 | 111 | **141** | **126** | **225** |

Table 2: Mean squared error (multiplied by $10^3$) for different MLP architectures across synthetic distributions. Lower is better.

| Architecture | Gaussian | | | Ring | Moons | Ind. mixture | Octagon |
|---|---|---|---|---|---|---|---|
| | 2d | 3d | 5d | | | | |
| d-40-1 | 1832 | 296 | 753 | 678 | 2124 | 957 | 1224 |
| d-20-20-1 | **99.1** | **192** | 583 | 355 | 1202 | 701 | 894 |
| d-40-40-1 | 178 | 296 | 389 | 199 | 488 | 429 | 727 |
| d-80-80-1 | 500 | 211 | **309** | **145** | 567 | 601 | 465 |
| d-20-20-20-1 | 178 | 195 | 461 | 222 | **421** | **416** | **402** |

Several observations can be made from these results. First, the shallow architecture with a single hidden layer (d-40-1) performs noticeably worse than all deeper networks across every distribution for both metrics, confirming that depth is beneficial for density estimation. Second, among the deeper architectures, the three-layer network d-20-20-20-1 achieves the best performance on the three most complex two-dimensional distributions (Moons, Independent Mixture, and Octagon) for both KL divergence and MSE, demonstrating the advantage of depth for capturing intricate multimodal structure. For the Ring distribution, the two-layer d-80-80-1 performs best across both metrics.

For the Gaussian distribution, performance varies with dimension and metric choice. In two dimensions, the d-20-20-1 network yields the lowest KL divergence and MSE. In three dimensions, the results are mixed: d-20-20-1 achieves the best MSE while d-20-20-20-1 achieves the best KL divergence. In five dimensions, the wider d-80-80-1 network achieves the best results for both metrics. This suggests that higher-dimensional problems benefit from increased model capacity, though the optimal architecture depends on both dimension and distribution complexity.

For the two-dimensional Gaussian distribution, a parametric estimator (fitting a bivariate normal distribution via maximum likelihood) achieved a KL divergence of $0.61 \times 10^{-3}$ and MSE of $0.012 \times 10^{-3}$. While the parametric estimator naturally outperforms all nonparametric methods for this simple case, the proposed MLP estimators are substantially closer to these optimal values than the classical nonparametric baselines shown in the following section.

## 5.2 Comparison with Baseline Methods

The proposed method is compared against histograms, KDE, and k-NN estimators. For each baseline method, multiple hyperparameter configurations were evaluated: histogram bin counts of 5, 10, 20, 30, and 40 per dimension; KDE with Gaussian, tophat, Epanechnikov, exponential, and cosine kernels (bandwidth $h = 0.05$); and k-NN with $k = 5, 10, 20, 50, 100$. For the proposed method, the best-performing MLP architecture from table 1 is used for each distribution.

Table 3 reports the best KL divergence (multiplied by $10^3$) achieved by each method on each distribution. For baseline methods, the hyperparameter that achieved this result is indicated in parentheses.

On all five distributions, the proposed MLP method achieves the lowest KL divergence among nonparametric estimators. The improvement is most pronounced on complex distributions such as Octagon Gaussian Mixture, where classical methods struggle to capture the intricate multimodal struc-

Table 3: Best KL divergence (multiplied by $10^3$) achieved by each method. For baseline methods, the hyperparameter yielding the best result is shown in parentheses. Lower is better.

| Method | Gaussian ($2d$) | Ring | Moons | Ind. mixture | Octagon |
|---|---|---|---|---|---|
| Histogram (bins) | 144 (10) | 269 (20) | 302 (30) | 353 (20) | 628 (30) |
| KDE (kernel) | 14.85 (gauss) | 120 (cosine) | 220 (cosine) | 132 (cosine) | 271 (epan) |
| k-NN ($k$) | 65.55 (20) | 223 (10) | 308 (5) | 152 (10) | 241 (10) |
| MLP | **8.34** | **76.07** | **141** | **126** | **225** |

ture. For the simple Gaussian distribution, the MLP estimator is competitive with the best KDE result, while for more complex distributions the gap widens substantially.

## 5.3 VISUALIZATION OF DENSITY ESTIMATES

Visual inspection of the estimated densities provides additional insight beyond numerical metrics. Figures 2, 3, and 4 show the true densities and their estimates by k-NN and the proposed MLP method for three representative distributions: Gaussian, Independent Gaussian Mixture, and Octagon Gaussian Mixture.

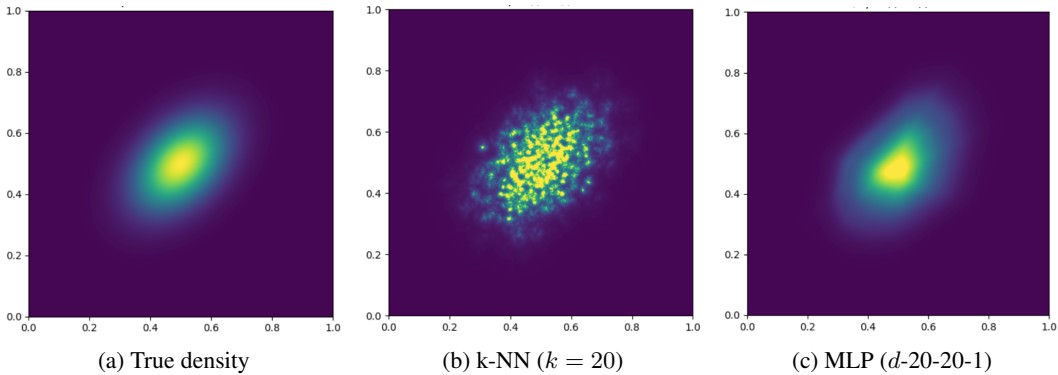

(a) True density      (b) k-NN ($k = 20$)      (c) MLP ($d$-20-20-1)

Figure 2: Density estimates for the Gaussian distribution.

For the Gaussian distribution (figure 2), all methods capture the overall unimodal structure. The k-NN estimate exhibits a characteristic "furry" appearance, with local fluctuations that arise from the discrete nature of nearest neighbor counting. The MLP estimate is smooth and visually closer to the true density, though it shows a slight asymmetry not present in the true distribution – an artifact of the specific training run.

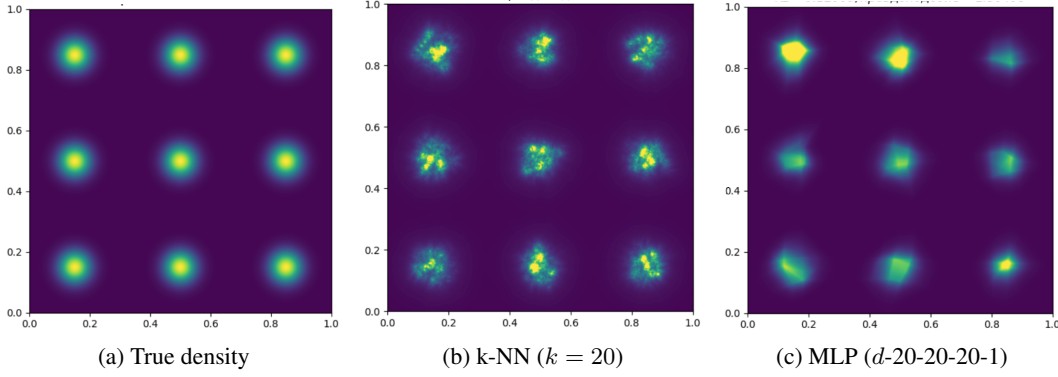

(a) True density      (b) k-NN ($k = 20$)      (c) MLP ($d$-20-20-20-1)

Figure 3: Density estimates for the Independent Gaussian Mixture.

The independent Gaussian mixture (figure 3) consists of nine modes arranged in a grid. The k-NN estimate reveals the multimodal structure but is noisy throughout. The MLP estimate resolves all

nine modes, but exhibits visible non-uniformity: some modes appear brighter and more pronounced than others, and the overall intensity varies across the grid. This reflects the difficulty of training the unary classifier to achieve perfectly balanced density estimates across all regions of the support.

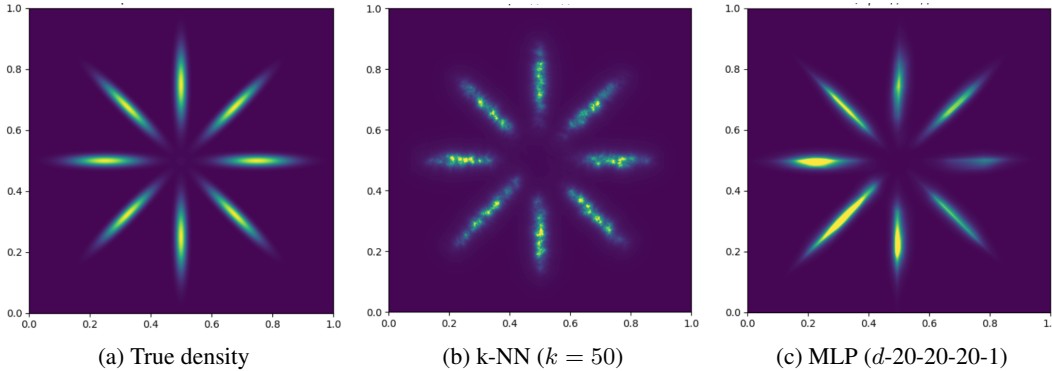

(a) True density     (b) k-NN ($k = 50$)     (c) MLP ($d$-20-20-20-1)

Figure 4: Density estimates for the Octagon Gaussian Mixture.

For the complex octagon Gaussian mixture (figure 4), the advantages and limitations of the proposed method are both evident. The true density consists of eight petals arranged in a circular pattern, with each petal elongated radially. The k-NN estimate reveals the petal structure but is extremely noisy, with the characteristic "furry" texture obscuring the true density contours. The MLP estimate faithfully reproduces the eight-petal structure and is visually much closer to the true density than k-NN. However, some non-uniformity is present: certain petals appear brighter and more intense than others, indicating that the density estimate is not perfectly balanced across all modes. This is a consequence of the training process – unary classification is inherently difficult to optimize, and the results presented here reflect a typical run rather than a carefully tuned best-case scenario.

Despite these imperfections, the MLP estimates are consistently smoother and more faithful to the true densities than their k-NN counterparts across all three distributions. The visual quality corresponds to the quantitative superiority reflected in the KL divergence measurements in table 3. The comparison demonstrates that the proposed method combines the adaptability of k-NN with the smoothness of kernel methods, while honestly reflecting the challenges of training neural networks for density estimation.

## 5.4 COMPUTATIONAL EFFICIENCY

Beyond estimation accuracy, computational efficiency at inference time is an important practical consideration. Table 4 reports the time required to evaluate each method on the two-dimensional evaluation grid (250000 points). All measurements were performed on CPU to ensure fair comparison, though the proposed MLP method could potentially be accelerated further on GPU hardware.

Table 4: Inference time (milliseconds) on a $500 \times 500$ grid. Ranges reflect different hyperparameter configurations.

| Method | Time (ms) |
|---|---|
| Histogram | 4.08 – 5.05 |
| MLP (proposed) | 4.78 – 72.0 |
| k-NN | 255 – 2212 |
| KDE | 1029 – 17343 |

The histogram is fastest but, as shown earlier, yields poor estimation quality. The proposed MLP is only slightly slower (depending on architecture) while delivering substantially better density estimates. In contrast, k-NN and KDE are orders of magnitude slower. This computational burden makes them impractical for real-time inference or repeated evaluations on large grids.

The MLP's efficiency stems from constant-time inference: once trained, each prediction requires a fixed number of operations regardless of training set size. This is a significant advantage over k-NN

and KDE, which scale linearly with the number of training points. The proposed method thus offers an attractive combination of accuracy and speed, with potential for further GPU acceleration.

## 6 Discussion

### 6.1 On KL Divergence Usage

A technical observation concerns KL divergence computation. Some studies normalize density estimates over a discrete grid, treating densities as probability mass functions – a mathematically incorrect practice that can yield misleading results, including negative values. In this work, KL divergence is computed via numerical integration over the continuous domain, ensuring proper accounting of density values.

### 6.2 Limitations

The method has several limitations. First, it suffers from the curse of dimensionality: as dimension increases, training points become sparse in the unit hypercube. Second, optimal network architecture varies across distributions, and selection without access to the true density remains an open problem.

### 6.3 Future Work

Future directions include adaptive noise generation for high-dimensional efficiency, extension to unbounded domains via different noise distributions (e.g., Gaussian), theoretical analysis of convergence rates, and application to real-world problems in anomaly detection and scientific data analysis, where interpretability is valuable.

## 7 Conclusion

This paper introduced a consistent nonparametric density estimator using neural networks. The method reframes density estimation as unary classification: an MLP with piecewise linear activations is trained to distinguish data from uniform noise, and its output is transformed into a density estimate via a simple formula.

The theoretical foundation rests on two results: adding background noise identifies the data support, and piecewise linear MLPs converge to histogram-based classifiers, ensuring consistency.

Experiments on synthetic distributions in two, three, and five dimensions demonstrate that the method consistently outperforms histograms, KDE, and k-NN in KL divergence. The improvement is most pronounced on complex multimodal densities, where classical methods struggle to capture intricate structure. Visualizations confirm that the MLP estimates faithfully reproduce the true density contours, despite some minor non-uniformity arising from the challenges of training unary classifiers.

Beyond accuracy, the method offers interpretability through its connection to adaptive histograms and efficient constant-time inference, unlike k-NN and KDE which scale with training set size.

Limitations include the curse of dimensionality, architecture selection, and training cost. Future work will address these through adaptive noise generation, extensions to unbounded domains, convergence rate analysis, and real-world applications.

The proposed estimator combines classical statistical guarantees with neural network flexibility, establishing a powerful tool for nonparametric density estimation.

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
